

# The impact of moderate-intensity basketball intervention on the physical self-esteem and school adjustment of first-year high school students

Wenting Wei[1,*], Ruirui Duan[2,*], Fulei Han[3] and Qiulin Wang[4]

[1] Kunming University, Kunming, Yunnan, China
[2] Changzhou Foreign Languages School, Changzhou, Jiangsu, China
[3] Kunshan No. 1 Vocational School in Jiangsu Province, Kunshan, Jiangsu, China
[4] Yangzhou University, Yangzhou, Jiangsu, China
[*] These authors contributed equally to this work.

## ABSTRACT

**Object**. This study conducted an 8-week experiment with the basketball sports intervention program to explore the relationship between the basketball sports intervention program and the physical self-esteem and school adjustment of first-year high school students, providing references for first-year high school students' school adjustment and related research.

**Methods**. Using convenient sampling,88 students in two classes of grade one in a senior high school in Changzhou were selected as the experimental research objects and were randomly divided into experimental groups (basketball) 44 people (20 male and 24 female; control group routine physical education) 44 people (23 male and 21 female). The exercise intervention lasted 8-week, 3-times-a-week with about 30 minutes of moderate-intensity exercise each time. The control group had normal sports activities, and the exercise intensity was not monitored. The Physical Self-esteem Scale and the School Adaptation Scale were used to measure the data in a group manner before and after the experiment. All data were statistically analyzed with SPSS26.0.

**Results**. (1) After 8 weeks of basketball intervention, the post-test level of school adaptation of first-year high school students was significantly higher than the pre-test ($p < 0.01$), and the post-test level of physical self-esteem was significantly higher than the pre-test ($p < 0.01$). (2) Basketball intervention can directly affect the school adaptation of first-year high school students and indirectly affect the school adaptation through physical self-esteem. The direct effect was 86.79%, and the indirect effect was 13.21%.

**Conclusion**. (1) Intervention of moderate intensity basketball can improve first-year high school students' school adaptation and physical self-esteem. (2) Intervention of moderate intensity basketball can directly improve first-year high school students' and school adaptation by improving individual physical self-esteem. Physical self-esteem plays a partial intermediary role between basketball and school adaptation.

Corresponding author
Qiulin Wang, wangql@yzu.edu.cn

## INTRODUCTION

In 2018, General Secretary Xi said at the National Education Conference: "We should adhere to the education concept of "health first", offer a variety of physical education classes, help students enjoy the fun of physical exercise, sharpen their will and strengthen their physical fitness". This points out the direction for the development of school sports in the future (*Shi, 2018*), the implementation of the improvement of students' physical and mental health, and the promotion of students' all-round development (*Ji, 2020*).

High school students are the future pillars of the motherland, and the high school period is also a critical stage in cultivating students' independence. In the process of development, puberty, the pressure of further education and academic pressure, and many other problems make high school students' mental health a broad concern for all walks of life (*Yang, 2018*). Under the present system in China, schools unilaterally pursue a high rate of advancement, parents' excessive expectations, overly focus on academic performance, and neglect the physical and mental health of high school students, especially the first year of high school students, which is highly unfavorable to their healthy growth (*Chen & Ji, 2006*). The natural social environment in high school is mainly the school environment, high-pressure classroom learning environment, and the external environment of students, teachers, and other groups, especially in the first year of high school, to the real world shows our school adaptation (*Song, 2017*). School is an essential place of learning for first-year high school students; therefore, good school adaptation benefits senior students' development. However, maladaptation will not only lead to communication difficulties, loneliness, depression, and a decline in academic performance; it will also significantly impact the future of work, marriage, and family (*Jiang, 2015*). Therefore, studying the school adaptation of first-year high school students is of far-reaching practical significance.

Preparing students for early integration into school life and enhancing school adjustment is one of the essential aims of education. Physical education as a part of education also has that role. Kim found that students who exercised regularly produced higher self-esteem levels, and achievement positively affected school adjustment (*Kim, Jang & Cho, 2020*). *Fitzgerald, Fitzgerald & Aherne (2012)* found that increased self-esteem through physical activity improved adjustment in adolescent girls. Other scholars found that moderate-intensity and higher-intensity exercise helped reduce symptoms related to school adjustment more than low-intensity exercise (*Conradsson et al., 2010*). Domestic scholars Peng Yeren and Chen Huina concluded that first-year college students's adjustment to school was significantly related to physical activity level and that students with a higher propensity to be physically active were more sensitive to the school life environment and interpersonal relationships compared with those with a lower propensity to be physically active (*Peng & Chen, 2007*). Through her study, *Cao et al. (2018)* found that physical activity can improve adolescents' school adjustment. However, research on exercise interventions and school adjustment has focused more on junior high school and college students, and there needs to be more research on exercise interventions for first-year high school students.

Self-esteem is at the core of mental health. Physical self-esteem is a specific domain of self-esteem that is effective in predicting the psychological benefits of exercisers (*Lou, 2017*), and improving physical self-esteem allows upper-level students to evaluate themselves better. Research has shown that students with high levels of physical self-esteem have higher levels of school adjustment than those with low physical self-esteem (*Jung, Kim & Kim, 2016*). Also, research has shown that exercise interventions can improve physical self-esteem (*Yang et al., 2014*).

Foreign scholars have found a meaningful correlation between obesity stress, self-esteem, and school adaptation, with physical activity reducing obesity levels and enhancing self-esteem and school adaptation (*Young & Young, 2014*). Scholars have found that participation in physical activity indirectly improves adolescents' school adjustment by increasing their self-perception (*De Moor et al., 2006*). Studies have also found that students who exercise regularly produce higher levels of self-esteem and achievement that positively impact school adjustment (*Kim, Jang & Cho, 2020*). *Fitzgerald, Fitzgerald & Aherne (2012)* found that increasing self-esteem through physical activity improved adjustment in adolescent girls. In conclusion, elevated levels of physical self-esteem enhance school adjustment, whereas decreased physical self-esteem increases the incidence of school adjustment distress. Therefore, physical activity improves school adjustment, possibly through improved physical self-esteem.

Basketball is a physically antagonistic sport that includes basketball technique and tactics, physical fitness, and quality (*Dong, 2017*), which is carried out exceptionally widely in China and is popular among high school students (*Zhu & Tang, 2021*). Studies have shown that basketball effectively promotes physical and mental health development and improves students' psychological quality. Basketball has been found to improve individual motor abilities, especially strength qualities and bouncing power, thus promoting health. (*Liu, 2017*). The combination of moderate-intensity basketball and figure skipping can indirectly enhance the self-confidence of upper primary school students by improving physical self-esteem (*Yan et al., 2019*). A study has found that basketball has an impact on students' school adjustment, improving their self-awareness as well as good interpersonal relationships at school in the process (*Liu, 2017*). Based on the above analyses, we take basketball as an experimental sport.

In summary, there are significant correlations between sports interventions and body self-esteem, sports interventions and school adjustment, and body self-esteem and school adjustment. However, studies have yet to find precisely how the three are related. Therefore, this study investigated whether body self-esteem plays a mediating variable in basketball sports intervention for school adjustment of first-year high school students.

A total of the following hypotheses were formulated in this study: (1) a moderate-intensity basketball intervention can improve school adjustment for first-year high school students; (2) a moderate-intensity basketball intervention can improve physical self-esteem for first-year high school students; (3) a moderate-intensity basketball intervention can indirectly improve school adjustment for first-year high school students by increasing the level of physical self-esteem.

## RESEARCH OBJECTS AND METHODS

### Respondents

The survey was conducted in a senior high school in Changzhou City, Jiangsu Province, where two classes were selected by convenience sampling method and randomly divided into experimental and control classes. An informed consent form was distributed to the respondents before the survey to obtain their understanding of and consent to complete the questionnaire.The ethics committee of the medical school of Yangzhou University approved the study (YXYLL-2023-153). A psychometric paper-and-pencil test was used, and uniform instruction was used to guide the subjects to fill out the questionnaire according to their situation. The questionnaires were collected on the spot. The sample size was estimated by F test using G*power 3.1 software. After calculation, a total sample size of 66 cases was required for both groups, and based on a 20% attrition rate, 79.2 cases were required for both groups, and 39.6 cases were required for each group. The sample size of this study was following the estimation. Each student completed the Physical Self-Esteem Scale and the School Adjustment Questionnaire. A total of 88 people agreed to fill out the questionnaires, and there were no invalid questionnaires, giving a valid sample of 88. The validity rate was 100 percent. The criteria for screening the questionnaire were: (1) complete information; (2) no multi-choice situation for one question; (3) no missed answers.

### Experimental method

Convenient sampling was used to extract students from a high school senior class in Changzhou City as experimental subjects, who were randomly divided into an experimental class and a control class. The students in the experimental group and the control group were all from senior one with an average age of 16.07 years. The experimental class was taught basketball according to the designed 8-week, three-times-a-week (Monday, Wednesday and Friday) basketball program; the control group was taught the 8-week, three-times-a-week (Monday, Wednesday and Friday) routine.According to the course standard, the experimental group set up ball practice, dribble, pass (including Quadrangle Pass) , shoot, tactical cooperation, game and other content.The control group set up the main teaching contents of track and field, including hurdle, long jump, throwing. Based on the high school student's curriculum syllabus and the observation results of favorite sports, combined with the actual situation of the school stadium, basketball is chosen as the content of the sports program. The experimental project is basketball, based on the general high school "physical education and health" curriculum standards; basketball characteristics require fast and powerful passing and catching, high accuracy, effortless control, dribbling sophisticated and skillful, stable shooting, and easy to combine with other actions. Skilled mastery of dribbling with different strengths and different speeds, different distances and different arcs of the pass and catch, and with shooting, can develop students' coordination ability, develop physical fitness to cultivate solidarity and cooperation between students, and joint development of physical fitness, enhance the purpose of the technology, and cultivate the students' courageous, decisive and good qualities. First-year high school students are strong in sensitivity and flexibility but poor in strength, quickly produce a sense of fatigue,

are lively and active, and are competitive. Therefore, in the design of teaching content, to achieve the objectives of this experiment to preset, with games and competitions to carry out a variety of teaching means, with a moderate amount of movement ball activities, and according to the situation of men and women and proficiency, set up learning group (teacher and apprentice grouping or male and female cooperation grouping), to achieve the purpose of the overall development of the student's physical fitness. Five randomly selected first-year high school students wore heart rate monitors and were monitored by assistants to ensure moderate exercise intensity. Quiet heart rate measurements were taken prior to the intervention, and subject heart rates were read according to time points and recorded throughout. Without avoiding the Hawthorne effect, this study adopted a single-masked experimental method, where the subjects in the experimental group were taught without knowledge of the experiment. In contrast, the control group was strictly controlled not to perform any physical activity related to basketball and the system.

### Physical self-esteem scale

The physical Self-Esteem Scale, compiled by *Xu & Yao (2001)*, is divided into five parts: motor skills, physical condition, physical attractiveness, physical fitness, and sense of self-worth. It has 30 questions, and the higher the score, the higher the level of physical self-esteem. In this study, Cronbach's alpha of the questionnaire was 0.9, Cronbach's alpha of the motor skills was 0.859, Cronbach's alpha of the Physical condition was 0.822, Cronbach's alpha of the physical attractiveness was 0.858, Cronbach's alpha of the physical quality was 0.848, Cronbach's alpha of the self worth was 0.850.

### School adaptation questionnaire

The School Adaptation Questionnaire, compiled by *Cui (2008)*, is divided into five parts: routine adaptation, learning adaptation, peer relationship, teacher–student relationship, and school attitude. It has 27 questions, and the higher the score, the higher the level of school adaptation. In this study, Cronbach's alpha of the questionnaire was 0.93, Cronbach's alpha of the School attitude was 0.851, Cronbach's alpha of the peer relationships was 0.870, Cronbach's alpha of the teacher-student relationship was 0.819, Cronbach's alpha of the academic adaptation was 0.871, Cronbach's alpha of the conventional adaptation was 0.871.

### Statistical methods

Statistical analyses were conducted using the social statistical analysis software SPSS 26.0, including homogeneity tests, repeated measures ANOVA, and path analysis mediation role modeling. Furthermore, $P < 0.05$ was considered statistically significant.

## RESULTS

### Control and inspection of standard method deviations

The Harman univariate test method was used to test whether a standard method bias existed in all project data. The results revealed that there were 15 factors with eigenvalues greater than 1, the maximum factor variance interpretation rate was 19.82%(<40%), and

**Table 1  Test of homogeneity between physical self-esteem and school adaptation.**

| Dimension | Control Group (N = 44) | Experimental Group (N = 44) | $t$ | $p$ |
|---|---|---|---|---|
| School attitude | 23.45 ± 6.673 | 24.77 ± 3.796 | −1.139 | 0.258 |
| Peer relationships | 24.55 ± 5.232 | 25.45 ± 3.682 | −0.943 | 0.349 |
| Teacher-student relationship | 16.91 ± 4.584 | 17.73 ± 2.790 | −1.011 | 0.315 |
| Academic adaptation | 16.14 ± 3.254 | 16.80 ± 2.882 | −1.006 | 0.317 |
| Conventional adaptation | 14.95 ± 3.457 | 15.27 ± 2.564 | −0.490 | 0.625 |
| School adaptation total score | 96.00 ± 17.337 | 100.02 ± 10.881 | −1.304 | 0.196 |
| Motor skills | 12.45 ± 4.184 | 12.68 ± 4.821 | −0.236 | 0.814 |
| Physical condition | 13.70 ± 4.787 | 13.75 ± 4.591 | −0.045 | 0.964 |
| Physical attractiveness | 13.05 ± 4.253 | 12.57 ± 3.631 | 0.566 | 0.573 |
| physical quality | 13.11 ± 3.919 | 13.89 ± 4.566 | −0.852 | 0.397 |
| Self worth | 12.77 ± 3.595 | 12.80 ± 4.289 | −0.027 | 0.979 |
| Physical self-esteem Total score | 65.09 ± 15.413 | 65.68 ± 18.125 | −0.165 | 0.870 |

no factors with excessive explanatory power. Thus, no standard severe method deviation problem existed in the data used in this study.

## Test of homogeneity between physical self-esteem and school adaptation

Homogeneity was tested using an independent samples $t$-test, which showed that the difference between physical self-esteem and school adjustment between the test and control groups was not statistically significant (Table 1).

## The effects of an exercise intervention on physical self-esteem and school adjustment in first-year high school students

In order to explore the effects of different time and group factors on the school adjustment of first-year high school students, this study adopted a two-factor repeated-measures ANOVA to analyze the data before and after the experiment. The results of the study showed (Table 2) that there was a main effect of the time factor [$F_{(1,86)} = 47.809$, $p < 0.01$, $\eta_p^2 = 0.526$], a main effect of the group factor [$F_{(1,86)} = 11.409$, $p < 0.01$, $\eta_p^2 = 0.210$], and an interaction between the time factor and the group factor [$F_{(1,86)} = 98.210$, $p < 0.01$, $\eta_p^2 = 0.695$]. Simple effects analyses showed (Table 3) that at the pre-test, the difference between the total school adjustment scores of first-year high school students in the basketball group (100.02 ± 10.881 points) and the control group of seniors (96.00 ± 17.337 points) was not statistically significant, $p = 0.196$. At the post-test, the total school adjustment scores of the seniors in the basketball group were higher than that of the control group of seniors, and the difference was statistically significance, $p < 0.01$; in the control group, the difference between the total school adjustment scores of first-year high school students (94.80 ± 17.949) and the pre-test (96.00 ± 17.337) was not statistically significant, $p = 0.750$; in the basketball intervention group, the total school adjustment scores of first-year high school students (112.57 ± 15.001) were higher than that of the pre-test (100.02 ± 10.881), with a statistically significant difference. The difference between
**Table 2  Repeated measures analysis of variance for the impact of basketball intervention on school adjustment in first-year high school students.**

| Dependent variable | Source of variation | $F$ | $p$ | $\eta_p^2$ |
|---|---|---|---|---|
| School adaptation total score | Group | 47.809 | 0.000 | 0.526 |
| | Time | 11.409 | 0.002 | 0.210 |
| | Group × Time | 98.210 | 0.000 | 0.695 |
| School attitude | Group | 18.664 | 0.000 | 0.303 |
| | Time | 9.635 | 0.003 | 0.183 |
| | Group × Time | 20.020 | 0.000 | 0.318 |
| Peer relationships | Group | 0.113 | 0.739 | 0.003 |
| | Time | 8.012 | 0.007 | 0.157 |
| | Group × Time | 13.650 | 0.001 | 0.241 |
| Teacher-student relationship | Group | 15.735 | 0.000 | 0.268 |
| | Time | 9.053 | 0.004 | 0.174 |
| | Group × Time | 16.148 | 0.000 | 0.273 |
| Academic adaptation | Group | 1.884 | 0.177 | 0.042 |
| | Time | 9.924 | 0.003 | 0.188 |
| | Group × Time | 13.125 | 0.001 | 0.234 |
| Conventional adaptation | Group | 9.155 | 0.004 | 0.176 |
| | Time | 4.756 | 0.035 | 0.100 |
| | Group × Time | 6.368 | 0.015 | 0.129 |

the scores was statistically significant, $p < 0.01$. The results showed that school adjustment of first-year high school students improved significantly after eight weeks of basketball lessons three times a week.

In order to explore the effects of different time and group factors on the physical self-esteem of first-year high school students, this study adopted a two-factor repeated measures ANOVA to analyze the data before and after the experiment. The results of the study showed (Table 4) that there was a main effect of the time factor [$F_{(1,86)} = 214.599$, $p<0.01$, $\eta_p^2 = 0.833$], there was no main effect of the group factor [$F_{(1,86)} = 2.372$, $p = 0.131$, $\eta_p^2 = 0.052$], and there was an interaction between the time factor and the group factor [$F_{(1,86)} = 47.290$, $p < 0.01$, $\eta_p^2 = 0.524$], and a simple effects analysis showed (Table 5) that at the pre-test, there was no statistically significant difference between the total physical self-esteem scores of first-year high school students in the basketball group ($65.68 \pm 18.125$) and the control group of seniors ($65.09 \pm 15.413$) with a $p = 0.870$; and at the post-test, the total physical self-esteem scores of seniors in the basketball group were higher than the total physical self-esteem scores of first-year high school students in the control group and the difference was statistically significant, $p < 0.01$; in the control group there was no statistically significant difference between the total physical self-esteem scores of first-year high school students ($70.00 \pm 15.832$) and the pre-test ($65.09 \pm 15.413$), $p = 0.144$; in the basketball exercise intervention group, the total physical self-esteem scores of first-year high school students ($80.48 \pm 16.145$) were higher than the The difference between the pre-test ($65.68 \pm 18.125$ points) was statistically significant, $p < 0.01$. The results showed

**Table 3  Pre and post-intervention values of school adaptation in first-year high school students for the experimental group and control group (M ± SD).**

| Dependent Variable | Experimental Group (N=44) | | Control Group (N=44) | | Pre-*P* | Post-*P* |
|---|---|---|---|---|---|---|
| | Pre-Intervention | Post-Intervention | Pre-Intervention | Post-Intervention | | |
| School adaptation total score | 100.02 ± 10.881 | 112.57 ± 15.001 | 96.00 ± 17.337 | 94.80 ± 17.949 | 0.196 | <0.01 |
| School attitude | 24.77 ± 3.796 | 28.95 ± 4.951 | 23.45 ± 6.673 | 23.68 ± 5.790 | 0.258 | <0.01 |
| Peer relationships | 25.45 ± 3.682 | 27.02 ± 3.358 | 24.55 ± 5.232 | 23.32 ± 5.148 | 0.349 | <0.01 |
| Teacher-student relationship | 17.73 ± 2.790 | 20.86 ± 3.508 | 16.91 ± 4.584 | 17.23 ± 4.812 | 0.315 | <0.01 |
| Academic adaptation | 16.80 ± 2.882 | 18.50 ± 3.638 | 16.14 ± 3.254 | 15.43 ± 3.399 | 0.317 | <0.01 |
| Conventional adaptation | 15.27 ± 2.564 | 17.23 ± 2.932 | 14.95 ± 3.457 | 15.14 ± 3.203 | 0.625 | <0.01 |

that the physical self-esteem of first-year high school students improved significantly after eight weeks of basketball lessons three times a week.

## Modeling the effects of a basketball intervention on school adjustment in first-year high school students

Regression analysis was carried out with the post-experimental measurements of school adaptation of senior students as the dependent variable and different groups as the independent variables, $F_{(1,86)} = 25.399$, $P < 0.01$, $R = 0.477$, $R^2 = 0.228$, $\Delta R^2 = 0.219$, which indicated that the regression equation was significant and that the basketball intervention had a highly significant effect on school adaptation of senior students.

Taking the scores of senior students' physical self-esteem post-experiment measurements as the dependent variable and different groups as the independent variable, regression analysis was conducted, $F_{(1,86)} = 9.446$, $P < 0.01$, $R = 0.315$, $R^2 = 0.099$, $\Delta R^2 = 0.088$, indicating that the regression equation is significant. The basketball intervention has a very significant effect on the physical self-esteem of senior students.

The regression analysis was carried out with the scores measured after the experiment on school adaptation of senior students as the dependent variable and with the scores measured after the experiment on physical self-esteem of different groups as the independent variable, and the regression analyses were carried out, and the results of $F_{(2,85)} = 4.153$, $P < 0.05$, $R = 0.514$, $R^2 = 0.264$, $\Delta R^2 = 0.247$ indicated that the regression equation was significant, and that the effect of basketball sports intervention on the physical self-esteem and school adaptation of senior students was very significant.

In summary, the mediating effect of body esteem was significant (Table 6). The mediating effect model is shown in Fig. 1. It shows that basketball sports intervention has a direct effect on senior students' school adjustment; on the other hand, basketball sports intervention has an indirect effect on senior students' school adjustment through body self-esteem. The mediating effect of body esteem as a percentage of the total effect was $0.315 \times 0.200/(0.477) \times 100\% = 13.21\%$.
**Table 4** Repeated measures analysis of variance for the impact of basketball intervention on physical self-esteem in first-year high school students.

| Dependent variable | Source of variation | $F$ | $p$ | $\eta_p^2$ |
|---|---|---|---|---|
| Physical self-esteem total score | Group | 214.599 | 0.000 | 0.833 |
| | Time | 2.372 | 0.131 | 0.052 |
| | Group × Time | 47.290 | 0.000 | 0.524 |
| Motor skills | Group | 42.865 | 0.000 | 0.499 |
| | Time | 2.249 | 0.141 | 0.050 |
| | Group × Time | 11.219 | 0.002 | 0.207 |
| Physical condition | Group | 23.354 | 0.000 | 0.352 |
| | Time | 1.646 | 0.206 | 0.037 |
| | Group × Time | 10.868 | 0.002 | 0.202 |
| Physical attractiveness | Group | 19.728 | 0.000 | 0.315 |
| | Time | 1.089 | 0.303 | 0.025 |
| | Group × Time | 9.954 | 0.003 | 0.188 |
| physical quality | Group | 46.407 | 0.000 | 0.519 |
| | Time | 2.528 | 0.119 | 0.056 |
| | Group × Time | 1.151 | 0.289 | 0.026 |
| Self worth | Group | 47.543 | 0.000 | 0.525 |
| | Time | 1.539 | 0.222 | 0.035 |
| | Group × Time | 9.496 | 0.004 | 0.181 |

**Table 5** Pre- and post-intervention values of physical self-esteem in first-year high school students for the experimental group and control group (M ± SD).

| Dependent Variable | Experimental Group (N=44) | | Control Group (N=44) | | Pre-$P$ | Post-$P$ |
|---|---|---|---|---|---|---|
| | Pre-Intervention | Post-Intervention | Pre-Intervention | Post-Intervention | | |
| Physical self-esteem total score | 65.68 ± 18.125 | 80.48 ± 16.145 | 65.09 ± 15.413 | 70.00 ± 15.832 | 0.870 | <0.01 |
| Motor skills | 12.68 ± 4.821 | 15.93 ± 4.283 | 12.45 ± 4.184 | 13.34 ± 4.126 | 0.814 | <0.05 |
| Physical condition | 13.75 ± 4.591 | 16.70 ± 4.354 | 13.70 ± 4.787 | 14.34 ± 4.092 | 0.964 | <0.05 |
| Physical attractiveness | 12.57 ± 3.631 | 15.30 ± 3.400 | 13.05 ± 4.253 | 13.39 ± 3.823 | 0.573 | <0.05 |
| physical quality | 13.89 ± 4.566 | 14.98 ± 3.593 | 13.11 ± 3.919 | 16.59 ± 3.775 | 0.397 | <0.05 |
| Self worth | 12.80 ± 4.289 | 15.95 ± 3.754 | 12.77 ± 3.595 | 13.95 ± 3.935 | 0.979 | <0.05 |

**Table 6** Mediation analysis.

| Step | Standardized regression equation | Regression coefficient test |
|---|---|---|
| Step 1 | $Y = 0.477x$ | SE = 3.527, $t = 5.040$** |
| Step 2 | $Y = 0.315x$ | SE = 3.409, $t = 3.073$** |
| Step 3 | $Y = 0.415x$ | SE = 3.649, $t = 4.230$** |
| | $+0.200z$ | SE = 0.110, $t = 2.038$* |

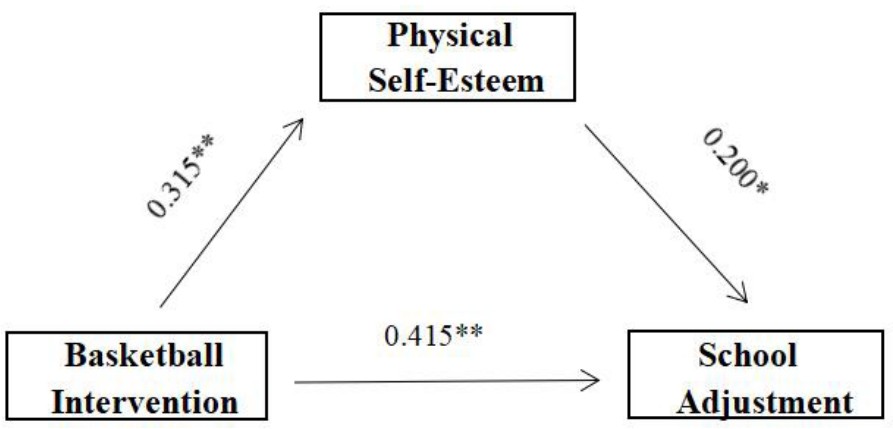

**Figure 1** Diagram of the mediation model.

## DISCUSSION

### The effects of a basketball intervention on upper-level students's school adjustment

The results of this study showed that after eight weeks of basketball exercise, the overall school adjustment of first-year high school students in the basketball exercise intervention group improved significantly. Likewise, the dimensions of school adjustment changed in a positive direction. Regarding school attitudes (Tables 2 and 3), first-year high school students continued to familiarise themselves with the school and adapt to the school environment during practice, thus increasing their sense of belonging and reliance on the school. Regarding peer relationships (Tables 2 and 3), basketball can increase communication between classmates and promote emotional warming. Good peer relationships can create positive emotional attitudes toward school among first-year high school students. At the same time, basketball can cultivate a sense of cooperation and teamwork among first-year high school students and promote positive emotional relationships among peers. In terms of teacher-student relationships (Tables 2 and 3), after eight weeks of living together, students and teachers have established deep friendships with each other, which promotes good teacher-student relationships, and first-year high school students can actively participate in class and school activities, which further promotes students' classroom performance in class, and enables them to learn basketball skills more quickly. Regarding academic adjustment (Tables 2 and 3), good interpersonal relationships, especially teacher-student relationships, can promote students' academic growth and active learning. In routine adaptation (Tables 2 and 3), first-year high school students could adapt quickly to returning to school after the epidemic and were fully engaged in their academic life. The study found that basketball is a collective sport, which can strengthen the cooperation and communication between students and require synergistic cooperation between individuals, and the passing and receiving, cover cooperation, and blocking techniques in basketball requires tacit cooperation between individuals and the ability to think differently from each other, which can strengthen the interaction and communication

between students and teachers, and can enhance the degree of intimacy between students and teachers. The study also found that good inter-school interpersonal relationships can promote school adaptation, including teacher-student relationships, school attitudes, and other aspects so that students deepen their emotions towards the school, which is conducive to better school adaptation (*Chen et al., 2019*; *Harris, Altekruse & Engels, 2003*; *Kliziene et al., 2018*; *Ren et al., 2017*; *Xiong, Liu & Zhang, 2020*). Compared with other studies, the results of this study are consistent with the performance of basketball team events in peer relationship, teacher-student relationship and school attitude dimensions, and the performance of this study is more obvious, the effects on academic adaptation and routine adaptation are less pronounced. Overall, performing basketball practice significantly affects students' school adjustment. Thus, hypothesis H1 is verified. And there is no main effect of time for the two dimensions of peer relationship and school adjustment, which indicates that peer relationship and school adjustment do not change much over time, and the scores of the pos *t*-test are lower than those of the pre-test (Tables 2 and 3), which may be due to the following reasons: firstly, the current intervention was opened after the improvement of the epidemic, and the students met each other more on the computer, with very little communication, coupled with the limited time of the intervention, which led to the existence of some; Secondly, as students who just entered high school from junior high school, their personalities changed compared to junior high school, and they paid more attention to themselves, with little communication among peers, and under the influence of the epidemic, communication among classmates was even more rare; then, under the influence of the epidemic, the time and place of students' classes were highly irregular, and they faced computers in class at any time, which led to unstable classes and affected their academic performance; finally, the regular physical education classes were slightly unstable. Finally, regular physical education classes could be more exciting. During the pandemic, students are asked to keep their distance and lack communication, which is different from the cooperative communication of basketball in the experimental group, in which many exercises require cooperation between two or more people, which is conducive to communication between each other.

## The effects of a basketball intervention on upper school students' physical self-esteem

The results of this study show that after eight weeks of basketball, the overall condition of physical self-esteem of first-year high school students in the basketball intervention group was significantly improved, which verified our initial research hypothesis and was also supported by related studies that physical education and sports have the effect of promoting the physical and mental health of senior students, and that some of the indicators of physical and mental health have a close correlation with the occurrence of physical self-esteem (*Bai, 2022*), and that similarly, physical self-esteem dimensions also changed in a positive direction, specifically in the four dimensions of motor skills, physical condition, physical attractiveness, and sense of self-worth, with a slight improvement in the physical fitness dimension, and a non-significant main effect of the group factor, suggesting that the differences between the different intervention groups were not significant. Regarding

motor skills (Tables 4 and 5), the ratings of the ability to learn new skills and the level of self-confidence were included. In basketball learning, there are strict requirements for first-year high school students' strength, coordination, speed, *etc.*; they need to maintain good body posture and control their core in the practice process. After eight weeks of basketball learning, first-year high school students' ability and skills in all sports were improved, and they could use their basketball skills, which stimulated the students' interest in sports and helped them to improve their self-confidence. In terms of physical fitness (Tables 4 and 5), first-year high school students improved their muscular, respiratory, and physical fitness levels through the coordinated development of their upper and lower limbs to complete basketball exercises. In terms of physical attractiveness (Tables 4 and 5), first-year high school students can discover their changes through this process of change, which can cultivate the beauty of healthy, positive, and optimistic physical form of senior students, thus promoting the improvement of self-confidence. In terms of physical self-worth (Tables 4 and 5), through continuous practice, first-year students continue to gain physical and psychological satisfaction, which promotes their appreciation of their bodies and enhances their confidence in their appearance, making them appreciate themselves more in life. Therefore, after eight weeks of basketball practice, physical self-worth has a positive impact. In terms of physical fitness (Tables 4 and 5), the duration of the training cycle needed to be longer, which led to less pronounced improvement and slower progress for the students. Compared with other studies, the results of this study are consistent with the performance of basketball team events in terms of motor skills, physical status, physical attractiveness and self-worth; however, there was only a small improvement in the fitness dimension, perhaps because of the short duration compared with other studies. Overall, conducting basketball practice has a significant effect on students' physical self-esteem. Thus, hypothesis H2 is verified. Practicing in basketball class makes students focus on their self-perception and identification of their sense of self-worth, especially students' improvement in motor skills, physical condition, physical attractiveness, and sense of self-worth is more prominent. The analysis of the experimental data (Tables 4 and 5) showed that the students' self-confidence in the experimental group increased significantly, and their interest in learning basketball increased. The recognition of basketball is high. This shows that basketball teaching has a positive effect on physical self-esteem and, even more so, on psychological benefits. At the same time, teachers were instructed to carry out targeted educational activities to dig deeper into the potential of students' basketball in the teaching process, personalize the teaching, and improve students' lifelong sports awareness.

### Physical self-esteem mediates the relationship between basketball intervention and school adjustment in upper school students

Through the analysis of the established structural equation model, it was learned that body self-esteem played a partially mediating role and had a significant effect on basketball sports intervention and school adaptation, indicating that basketball sports intervention could be achieved through the indirect effect of body self-esteem in addition to its direct effect on school adaptation. Research has shown that physical activity reduces obesity and enhances self-esteem and school adjustment (*Young & Young, 2014*). Scholars have

found that participation in physical activity can indirectly improve adolescents' school adjustment by improving their self-perception (*De Moor et al., 2006*). Studies have also found that physical self-esteem is predictive of school adjustment in high school students (*Angelo Junior et al., 2021*; *Sung-Hoon & Jong-Hoon, 2017*) and that students who exercise regularly produce higher levels of self-esteem and achievement have a positive impact on school adjustment (*Kim, Jang & Cho, 2020*). *Fitzgerald, Fitzgerald & Aherne (2012)* found that increasing self-esteem through physical activity improves adjustment in adolescent girls. The present study found that increasing students' physical self-esteem may improve school adjustment. In physical education classes, especially basketball, which is a physically confrontational sport, extra attention is paid to physical training and motor skills, and the process of their rising athletic level is bound to be accompanied by an increase in the level of self-esteem of the body, leading to a reasonable school adjustment of the students. Therefore, if we want to improve the school adjustment of first-year high school students, it is essential to improve the body's self-esteem.

### Shortcomings of this study and prospects

Due to the impact of the epidemic, the original 12-week experiment was shortened to 8 weeks, which is a relatively short span; in the future, consideration should be given to resuming the 12-week or increasing the intervention time to 16 weeks and adding intermediate measurements in the course of the experiment; the mediator was only considered to be an indicator of self-esteem, but in the future, more indicators can be added to carry out the research; a cross-lagged model can be set up with the three variables of physical exercise, self-esteem, and school adjustment to conduct follow-up studies on the subjects.

## CONCLUSION

(1) A moderate-intensity basketball intervention can improve school adjustment for first-year high school students; (2) a moderate-intensity basketball intervention can improve physical self-esteem for first-year high school students; (3) a moderate-intensity basketball intervention can indirectly improve school adjustment for first-year high school students by increasing the level of physical self-esteem.

### Funding

This research was funded by the 2022 Ministry of Education Humanities and Social Research Project (Grant number: 22YJE890001). The funders had no role in study design, data collection and analysis, decision to publish, or preparation of the manuscript.

### Grant Disclosures

The following grant information was disclosed by the authors:
2022 Ministry of Education Humanities and Social Research Project: 22YJE890001.

## Competing Interests

The authors declare there are no competing interests.

## Author Contributions

- Wenting Wei conceived and designed the experiments, performed the experiments, authored or reviewed drafts of the article, and approved the final draft.
- Ruirui Duan conceived and designed the experiments, performed the experiments, analyzed the data, prepared figures and/or tables, and approved the final draft.
- Fulei Han conceived and designed the experiments, performed the experiments, analyzed the data, prepared figures and/or tables, authored or reviewed drafts of the article, and approved the final draft.
- Qiulin Wang analyzed the data, authored or reviewed drafts of the article, and approved the final draft.

## Human Ethics

The following information was supplied relating to ethical approvals (*i.e.*, approving body and any reference numbers):

The ethics committee of the medical school of Yangzhou University approved the study (YXYLL-2023-153).

## Data Availability

Raw data is available in the Supplemental Files.

## Supplemental Information

Supplemental information for this article can be found online at http://dx.doi.org/10.7717/peerj.17941#supplemental-information.

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
