# Peer review of "The impact of moderate-intensity basketball intervention on the physical self-esteem and school adjustment of first-year high school students"

_PeerJ, doi:10.7717/peerj.17941_

## Round 0.1 · original submission · Minor Revisions

The manuscript has potential and provides interesting data. In order to be publishable all points raised by both reviewers should be addressed.

Reviewer 1 ·

Basic reporting

the paper adresses an interesting subject within the basketball field.
in the introduction section the phrase in the quotation marks, because its a direct citation of the authors, should have the page number form where it was referenced.
in the last paragraph of the introduction section, before the hypotheses should be a very well defined and specific objetive of the study, so that in order to pursuit that purpose the hypotheses are presented.
the main introduction is very good.

Experimental design

the authors should add in the methods section, in which time period of the day were the evaluations performed... all in the morning or afternoon??
how and wich instrument was used for heart rate measurements?

Validity of the findings

all the findings are very well represented and clear. however i encourage authors to add in tables 3 and 5 the p values of the comparisons between pre and post tests.

Additional comments

no comments

Reviewer 2 ·

Basic reporting

The manuscript investigates an interesting and current topic. The introduction provides relevant information related to the role of physical activities on physical self-esteem and school adjustment for teenagers in the first year of high school. References are relevant and consistent with the ideas supported (especially for the introduction section). The importance of basketball for Chinese youth, the effects on psychological, social, fitness level and health are identified. The results are well summarized in tables and analyzed in the discussion section.

Experimental design

At the end of the introduction, the authors clearly indicated the three directions of the research. The two applied questionnaires are consistent with the investigated theme and provide solidity to the study. The duration of the experiment and the number of weekly sessions are specified. The authors have attached additional documents, indicating the individual values of the participants/Excel Table, the consent agreement and the approval of the Ethics Committee for the conduct of the research. The applied statistical procedures ensure a detailed exploration of the research topic.

Validity of the findings

The results summarized in the tables indicate the mean values at the initial and final tests of the experimental and control groups, for the factors of the two questionnaires/dependent variables. F, sig. and effect size values for the independent variables Group, Time and for the Group*Time interaction are shown. The conclusions are well correlated with the research directions.

Additional comments

1. Did all students in the experimental group/sample have continuity in basketball instruction (ie, did they participate in physical education lessons with basketball assignments in the previous/middle school cycle)? I think that there can be major difficulties related to learning and adaptation, if they have practiced another team sports game (soccer, volleyball, handball...).
2. It is not clear what the physical activity of the control group consisted of. They did not have basketball activities, but it would be useful to specify what curriculum content was taught in the 3 activities planned weekly x 8 weeks. Thus, the usefulness and efficiency of the basketball game (confirmed by the statistical data presented) would be clearer, if you indicated with which other types of physical activity it is compared.
3. Are there differences between females and males for the mean values of the questionnaire items? You could use this data in other scientific publications.
4. There is a lack of information related to the age of the studied samples and the exact period/time interval in which you applied the experimental basketball content curriculum.
5. It is not clear what the heart rate values were for efforts to fall into the zone of moderate demands/efforts. The game of basketball is very dynamic and there can also be very high functional demands, where energy is provided by the anaerobic system.
6. You used two questionnaires (with 30 and 27 items, respectively). Each of these questionnaires has 5 factors/dimensions/components, analyzed in tables as dependent variables. Cronbach's alpha values are specified in the text only at the level of the entire questionnaire. It might be useful to also indicate the separate values for each component (only if these values have already been calculated).
7. Discussion Section: (The Effects of a Basketball Intervention on upper-level students' school adjustment and The Effects of a Basketball Intervention on Upper School Students' Physical Self-Esteem). Perhaps it would be useful to add a short paragraph in which you compare the results obtained with other similar research. There are only a few references analyzed (lines 319-320 and 344).

---

## Round 0.2 · accepted · Accept

I am pleased to inform you that both reviewers and I agree with the edits. The manuscript is now ready for publication-

Reviewer 1 ·

Basic reporting

the authors adressed all changes as requested.

Experimental design

the authors adressed all changes as requested.

Validity of the findings

the authors adressed all changes as requested.

Additional comments

the authors adressed all changes as requested.

Reviewer 2 ·

Basic reporting

There are no additional comments.

Experimental design

There are no additional comments.

Validity of the findings

There are no additional comments.

Additional comments

The authors responded favorably to all the suggestions sent to the first evaluation of the manuscript.
Congratulations!